# Utilizing Molecular Simulations to Examine Nanosuspension Stability

Andrew P. Latham [1,2,*,†] , Elizabeth S. Levy [1], Benjamin D. Sellers [2] and Dennis H. Leung [1,*]

1. Small Molecule Pharmaceutical Sciences, Genentech, Inc., 1 DNA Way, South San Francisco, CA 94080, USA
2. Discovery Chemistry, Genentech, Inc., 1 DNA Way, South San Francisco, CA 94080, USA
* Correspondence: alatham30@gmail.com (A.P.L.); leung.dennis@gene.com (D.H.L.)
† Current address: Department of Bioengineering and Therapeutic Sciences, Quantitative Biosciences Institute, University of California, San Francisco, San Francisco, CA 94143, USA.

**Abstract:** Drug nanosuspensions offer a promising approach to improve bioavailability for poorly soluble drug candidates. Such formulations often necessitate the inclusion of an excipient to stabilize the drug nanoparticles. However, the rationale for the choice of the correct excipient for a given drug candidate remains unclear. To gain molecular insight into formulation design, this work first utilizes a molecular dynamics simulation to computationally investigate drug–excipient interactions for a number of combinations that have been previously studied experimentally. We find that hydrophobic interactions drive excipient adsorption to drug nanoparticles and that the fraction of polar surface area serves as a predictor for experimental measurements of nanosuspension stability. To test these ideas prospectively, we applied our model to an uncharacterized drug compound, GDC-0810. Our simulations predicted that a salt form of GDC-0810 would lead to more stable nanosuspensions than the neutral form; therefore, we tested the stability of salt GDC-0810 nanosuspensions and found that the salt form readily formed nanosuspensions even without the excipient. To avoid computationally expensive simulations in the future, we extended our model by showing that simple, two-dimensional properties of single drug molecules can be used to rationalize nanosuspension designs without simulations. In all, our work demonstrates how computational tools can provide molecular insight into drug–excipient interactions and aid in rational formulation design.

**Keywords:** formulation; nanosuspension; molecular dynamics

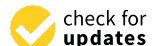



## 1. Introduction

Previous estimates suggest that approximately 40% of active pharmaceutical ingredients are poorly soluble in water [1]. To improve the bioavailability of such compounds, many formulations utilize drug nanosuspensions. Nanosuspensions possess a high ratio of surface area to volume, which greatly aids drug dissolution and increases saturation solubility [2–12]. Furthermore, they can be formed with minimal amounts of potentially toxic solvents and can be delivered through oral [13], dermal [14], pulmonary [15], or intravenous administration [16], which demonstrates both safety and convenience [17].

Such nanosuspensions can be formed through two classes of methods, broadly considered to be top-down methods and bottom-up methods. Top-down methods utilize wet milling to produce nanoparticles from larger drug crystals [18–21], while bottom-up methods precipitate nanocrystals out of a drug solution [22–24]. In both cases, it is often necessary to stabilize the nanosuspension through the use of excipients. These excipients adsorb to the drug crystal and prevent aggregation or Ostwald ripening, which can degrade the nanosuspension's stability [25]. Drug–excipient interactions are highly specific, and different excipients work better for different drugs; however, discovering the optimal excipient is often carried out through experimental screening [26–31]. Gaining insight into the interactions that drive excipient adsorption to drug nanoparticles could help rationalize formulation design.

Computational modeling offers a promising approach for examining the interactions that dictate nanosuspension stability [32–34]. Electronic structure methods such as density functional theory can be used to most accurately evaluate drug–nanoparticle interactions [35,36]. At the cost of some accuracy relative to electronic structure methods, molecular dynamics (MD) simulations can study larger systems and explore longer timescales, making them advantageous in a variety of drug nanosuspension studies. Indeed, MD simulations have been used to calculate the rate of dissolution of different drug crystal polymorphs [37], determine the drug–excipient binding free energy [38], and examine the structural features of excipient adsorption to a drug [39–44]. The results of these studies suggest that computational modeling may provide crucial insight into how excipient–drug interactions dictate nanosuspension stability.

The aim of this work was to utilize MD simulations to elucidate the structural features that dictate nanosuspension stability. Previous work from Ferrar et al. synthesized drug nanoparticles via a top-down milling technique, characterized their stability through dynamic light scattering experiments, and noted that excipient amphiphilicity seemed important for nanosuspension stabilization [39]. To rationalize these previous experimental results, we created a computational framework to systematically investigate the same drug–excipient combinations. We performed MD simulations on each drug crystal and searched for properties predictive of experimental measurements of nanoparticle stability. After finding the polar fraction of solvent-accessible surface area predicts nanosuspension stability, we applied our predictions to a system not previously studied, GDC-0810, which is a former breast cancer candidate compound, is strongly hydrophobic and poorly soluble, and can be crystallized in isoforms with different polarities. This system served as an independent test of our model. We then used our library of simulated drug–excipient pairs to create a simplified, analytical model capable of making similar predictions without computationally expensive simulations. Overall, our work connects a microscopic measurement, simulated nanosuspension polarity, to a macroscopic quantity, experimental nanosuspension stability, and thus creates a pathway for rational formulation design.

## 2. Methods

### 2.1. Molecular Dynamics Simulation

To examine nanosuspension stability, all-atom molecular dynamics simulations were performed on a training set of three different drug molecules (naproxen, indomethacin, and itraconazole) with six different excipients (sodium dodecyl sulfate (SDS), sodium octyl sulfate (SOS), sodium deoxycholate (SDC), polyethylene glycol (PEG), polypropylene glycol (PPG), and poloxamer), as characterized by previous experiments reported in Ferrar et al. and in Tables S1–S3 [39]. After characterizing our training set, we then performed simulations with two different crystal forms of GDC-0810 [45], one of which is neutral and one of which is ionic. The GDC-0810 test simulations were performed with all six excipients that were utilized in the training set and Tween 80 for a total of seven excipients. Each simulation began by identifying the crystal structure of the drug. Naproxen (ID = COYRUD), indomethacin (ID = INDMET), and itraconazole (ID = TEHZIP) were extracted from the Cambridge Structural Database [46], while the GDC-0810 crystal structure was determined experimentally in a neutral (ID = gened) and ionic (ID = genea) form [45]. Each crystal was then replicated to a cube of approximately 35 Å per side using the periodic boundary conditions builder function in MOE 2020.9 [47]. Next, GROMACS 2020.4 editconf was used to center the crystal in a square simulation box with side lengths of 110 Å. Excipients were added randomly in solution to the simulation box using GROMACS insert-molecules [48]. The number of excipients added was chosen to match experiments:drug (wt.%) of 10%, and the exact number of drugs and excipients used in our simulations are provided for the training set (Table S4) and test set (Table S5).

Systems built by GROMACS were then loaded into Schrödinger 2020.4 Maestro [49]. Similar to the experimental formulations, we prepared computational drug–excipient combinations in aqueous solutions. This preparation was performed using Desmond

Simulation Builder, which solvated the simulation box with a side length of 110 Å with simple point charge (SPC) water, added sodium ions when they were necessary to balance the charge of excipients, and prepared the simulation to run in the OPLS3e force field [50]. Each setup was relaxed using the Desmond Minimization protocol, which performed 100 ps of Brownian dynamics at 10 K with a 1 fs timestep. We then ran production Desmond Molecular Dynamics simulations [51] under constant numbers, particles, and temperatures (NPT ensemble) for 200 ns at 300 K and 1.01325 bar. Restraints were placed on the drug crystal with a force constant of 5 kcal mol$^{-1}$ Å$^{-2}$. Data were saved every 100 ps, and the first 100 ns were excluded from all analysis for equilibration.

### 2.2. Simulation Analysis

Analysis of each simulation began by calculating a molecular contact matrix using MDAnalysis 1.1.1 [52,53]. Contacts between two molecules, either drug or excipient, were included if any heavy atoms were within 4.5 Å. This contact matrix then served three purposes: (1) we computed the atomistic contact maps, or frequencies at which excipient atoms and drug atoms were in contact; (2) we computed the probability distribution of whole molecule contacts; and (3) we output PDB files of the drug crystal and all excipients or counterions that were in contact with the drug crystal, which thus excluded water and any excipient or counterion not in contact with the crystal. We then input these PDB files into MOE 2020.09 to calculate the surface properties of our excipient–drug complex, including the fraction of polar surface area (FASA$_p$), accessible surface area (ASA), radius of gyration (R$_g$), and dipole. Fitting to a linear regression and least absolute shrinkage and selection operator (LASSO) were performed using MATLAB 2019b [54]. After calculating each initial linear regression model using all data available, we predicted its accuracy using k-fold cross validation. We repeated the k-fold division of our data 100 times and report the average and standard deviation of the classification accuracy for the data excluded from model fitting. Molecular properties of drugs and excipients were calculated using sdfCalcProps.csh, a script available through the chemalot open source package [55].

### 2.3. Flash Nanoprecipitation of GDC-0810

GDC-0810 is poorly soluble in aqueous solution but is ionizable at high pH, which improves its solubility. As such, we utilized flash nanoprecipitation (FNP), so that we could control pH throughout the formulation process. Nanoparticles with a diameter of 225 nm were prepared through FNP with a multi-inlet vortex mixer (MIVM) as previously described [56]. GDC-0810 was dissolved at 15 mg/mL in 0.4 eq NaOH, and 1 mL was added to a 1 mL syringe. An equal volume of water was added to 3 additional 1 mL syringes. Syringes were placed on the FNP, and the plate was compressed at an equal rate across the syringes. After passing through the MIVM, the nanoparticles entered a dilution chamber of 14 mL of water. The final solution was an 18-fold dilution, in water, of the original solution, which consisted of 15 mg/mL GDC-0810 in 0.4 eq NaOH. This process resulted in a final pH of 8.35. No organic solvents, surfactant, or stabilizer was used.

Nanoparticles were characterized by dynamic light scattering (Wyatt DynaPro Plate Reader III). Samples were diluted 10 fold; then, 30 µL was placed in a 384 black wall, clear bottom polystyrene microplate (Corning). Particle diameter by intensity was reported from 15 acquisitions.

## 3. Results

### 3.1. Hydrophobic Interactions Drive Excipient Adsorption to Drug Crystals

While nanosuspensions present a promising approach to improve the solubility and dissolution of active pharmaceutical ingredients, choosing the correct formulation to stabilize a chosen drug remains a difficult task. Traditional approaches have focused on more experimental characterization methods, but recent efforts have shown computational modeling may aid in the design of stable nanosuspensions [32–34]. To continue in this direction,

we applied molecular dynamics simulations to quantify drug–excipient interactions and find underlying physical characteristics that can predict nanosuspension stability.

We started by characterizing a set of systems studied previously by Ferrar et al. [39]. These previous experiments had characterized the nanoparticle size for three drug molecules (naproxen, indomethacin, and itraconazole) with six different excipients (sodium dodecyl sulfate (SDS), sodium octyl sulfate (SOS), sodium deoxycholate (SDC), polyethylene glycol (PEG), polypropylene glycol (PPG), and poloxamer). For each of these 18 possible drug–excipient combinations, we performed molecular dynamics simulations. For each simulation, a drug crystal was placed at the center of an aqueous simulation box, with excipient molecules dispersed randomly throughout the aqueous phase. Over time, the excipients adsorb to the drug crystal (Figure 1), with differences being observed for each drug–excipient pair (Figure S1).

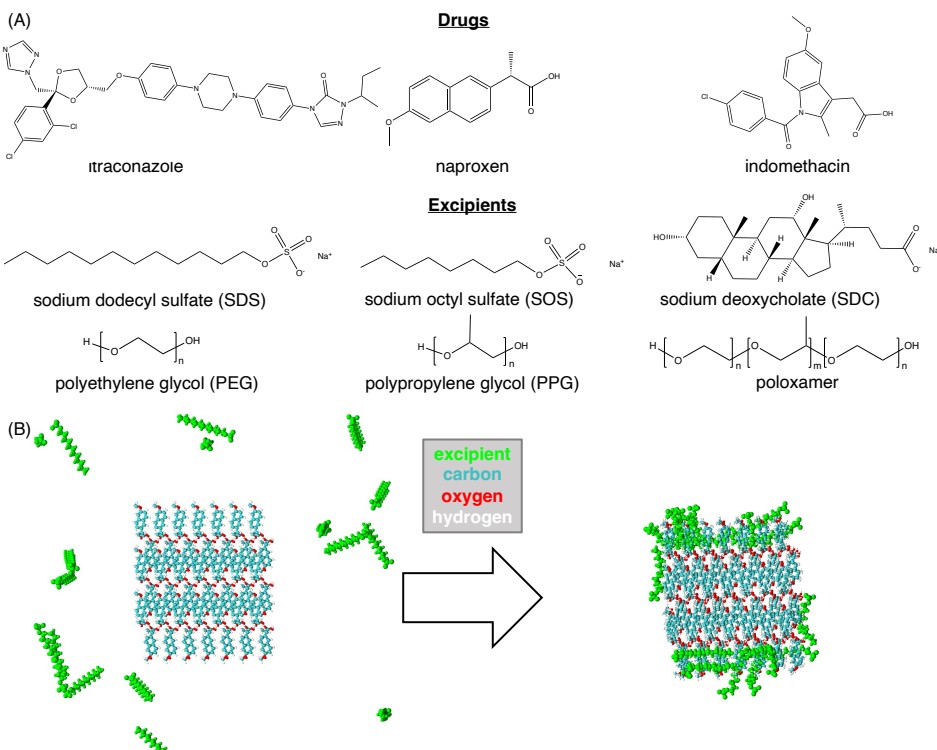

**Figure 1.** Summary of the primary training dataset used in this study. (**A**) Molecular structures are shown for each of 3 drugs and 6 excipients. For each drug, each of the 6 excipients were independently simulated for a total of 18 different simulations. In simulations of PEG and PPG, n = 20. In simulations of poloxamer, n = 5 and m = 10, which gives the same length as the PEG and PPG used in the simulation. (**B**) An example of our simulation methodology, with images taken from the start (left) and end (right) of our simulation of a naproxen crystal with SDS. Water molecules are hidden for improved clarity. At the start, excipients (green) were placed randomly around the drug crystal (C–blue, O–red, and H–white). Over the course of the simulation, the majority of the excipients adsorb to the drug crystal.

To quantify the specific interactions between drugs and excipients, we computed drug–excipient contact maps for naproxen (Figure 2), indomethacin (Figure S2), and itraconazole (Figure S3). These atomistic contact maps give insight into the parts of the drug and excipient that are most likely and least likely to form contacts between the drug and excipient. In all cases, we find that excipient–drug interactions are driven by hydrophobicity. As a result, drugs are contacted through exposed hydrophobic areas, such as the aromatic rings on naproxen and indomethacin or the terminal alkane group of itraconazole. Furthermore, evidence of hydrophobicity-driven interactions can also be seen on the excipients. Alkane

groups of SDS and SOS are frequently in contact with the drug particle, but the sulfate group is exposed to water. Similarly, SDC binds primarily through cyclohexane groups but exposes alcohol and carboxylic acid groups. Finally, polymers such as PEG, PPG, and poloxamer interact preferentially through carbon atoms, while oxygen atoms are less likely to form contacts with the drug crystal. Our simulations also show that the middle of the polymer has less contacts with the crystal than the ends of the chain, likely as a result of self-association or interactions with water. Viewed in total, these contact maps show that excipients interact with drugs through hydrophobic regions and leave more polar portions of the excipient or drug exposed.

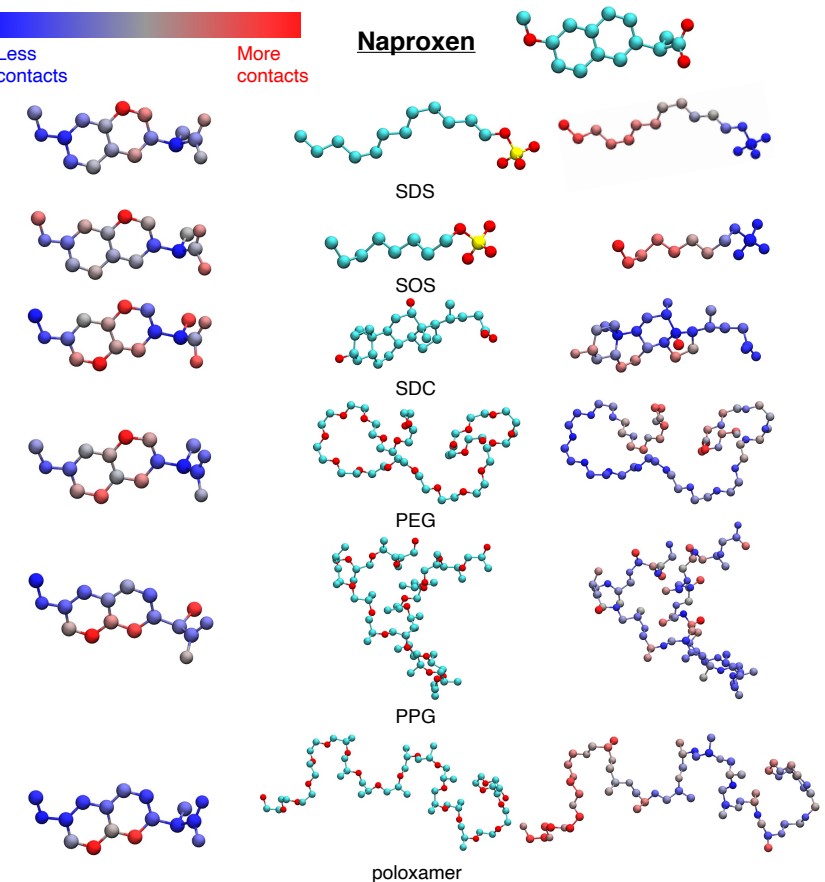

**Figure 2.** Heavy atom contact maps for where naproxen molecules come in contact with excipient molecules. For each excipient, we show a naproxen structure colored by the contact map (**left**), a structure of the excipient colored by atom type (**middle**), and a structure of the excipient colored by the contact map (**right**). For structures colored by atom type, the color mapping is C–cyan, O–red, and S–yellow. For structures colored by the contact map, red regions represent where the drug and excipient are in contact the most, while blue regions represent where the drug and excipient are in contact the least.

While atomistic pictures of these contact probabilities provides an interesting example of drug–excipient interactions, we can also quantify interactions between the excipient and drug nanocrystal in terms of molecule-to-molecule contacts or coordination number. These molecular contacts describe how many neighboring molecules of a given type a reference molecule may have. By examining the frequency of the number of drug molecules per excipient (Figure 3), the number of drug molecules per drug (Figure S4), the number of excipient molecules per drug (Figure S5), and the number of excipient molecules per excipient (Figure S6), we can better understand the ways in which drugs and excipients interact. We divide the data into surfactants (SDS, SOS, and SDC) and polymers (PEG, PPG,

and poloxamer) because the polymers have a larger molecular weight and, therefore, have a higher number of drug molecules per excipient.

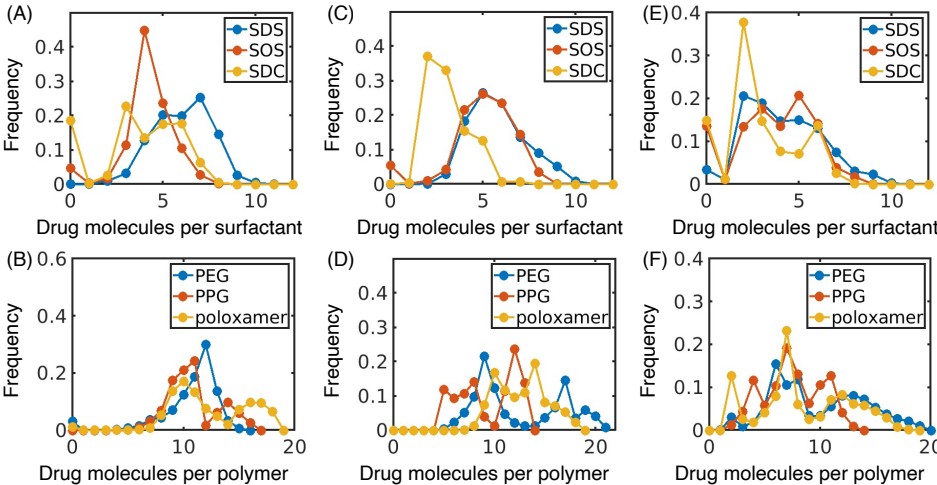

**Figure 3.** Frequency of the number of drug molecules in the nanocrystal bound to a single excipient molecule during our simulations. Distributions are broken up between surfactant and polymer excipients for naproxen (**A**,**B**), indomethacin (**C**,**D**), and itraconazole (**E**,**F**).

The number of drug molecules per surfactant reveals trends in the binding modes of surfactants with drug molecules (Figure 3A,C,E). In particular, SDC generally has a lower average coordination number than SOS or SDS. Previous simulations observed that the alkyl chains of SOS and SDS can penetrate the surface of the drug crystal, while SDC tends to lie on the surface of the crystal [39]. This mechanism of adsorption likely results in less drugs bound to SDC than SOS or SDS. Further, SOS and SDC frequently have 0 drug molecules per surfactant. This observation indicates that SOS and SDC have a lower affinity for the drug molecule than the more hydrophobic SDS, which results in SOS and SDC molecules that do not adsorb to the drug crystal.

The number of drug molecules per polymer also demonstrates the role of excipient hydrophobicity in adsorption. In comparison to the other polymers, PPG tends to have a lower number of drug molecules per polymer (Figure 3B,D,F) and higher number of polymer molecules per polymer (Figure S6). This indicates that being partially soluble may be important for a polymer to prevent excipient self-association and allow the polymer to interact with a higher fraction of the drug crystal. Taken together with the surfactants, these trends demonstrate that the interactions between drug and excipient are intricately controlled by balancing the excipient's affinity for drug, the excipient itself, and the aqueous phase.

### 3.2. Atomistic Modeling of Nanosuspension Stability

After examining how excipients interact with drug molecules, we aimed to connect our simulations to experimental nanosuspension stability by analyzing the excipient–drug complex. To accomplish this task, we isolated the drug crystal and all excipients and ions in contact with the crystal and calculated the accessible surface area (ASA), radius of gyration ($R_g$), fraction of the total surface area that is polar ($FASA_p$), and dipole moment for each drug–excipient pair (Figures 4A–C and S7A). We normalized ASA and $R_g$ according to that of the original crystal to remove effects arising from the differences in size of each drug crystal.

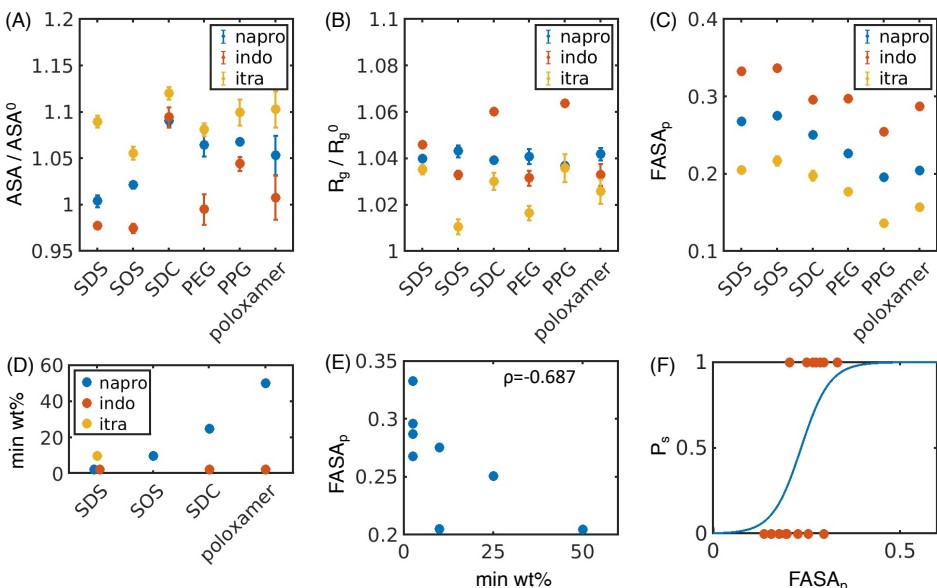

**Figure 4.** Utilizing surface descriptors to predict nanosuspension stability. (**A**) Accessible surface area of the excipient–drug system (ASA) divided by the accessible surface area of the drug crystal without excipients (ASA$^0$) for naproxen (napro, blue), indomethacin (indo, orange), and itraconazole (itra, yellow), with all excipients tested. (**B**) Radius of gyration of the excipient–drug system (R$_g$) divided by the radius of gyration of the drug crystal without excipients (R$_g^0$) for all drug–excipient combinations. (**C**) Fraction of polar surface area (FASA$_p$) for all drug–excipient combinations. (**A–C**) Error bars represent the standard deviation of estimates from five independent time windows. (**D**) Minimum weight percentage of excipient needed to form a stable nanosuspension (min wt.%) for each system that forms a stable nanosuspension. Data taken from Table S1 and Ferrar 2020 [39] and are listed explicitly in Table S3. (**E**) Fraction of polar surface area (FASA$_p$) plotted against the minimum weight percentage of excipient needed to form a stable nanosuspension (min wt.%) for each drug–excipient pair. The Pearson correlation coefficient ($\rho$) between FASA$_p$ and min wt.% is also shown. (**F**) Using FASA$_p$ to predict the probability that a stable nanosuspension is formed ($P_s$) for each drug–excipient pair using a logistic regression. True data (orange dots) and the resulting fit (blue line) are both shown.

With various simulated descriptors in hand, we then aimed to compare our results to experimental measures of nanosuspension stability. As an ideal formulation will minimize the ratio of excipient:drug, we used the minimum weight percentage of excipient needed to form a stable nanosuspension (min wt.%) as our target metric and performed additional experiments to complement the dataset from Ferrar et al. (Table S1) [39]. Of the 16 test drug–excipient combinations, 8 formed stable nanosuspensions, for which we recorded the min wt.% (Figure 4D, Table S2). Next, we calculated the Pearson correlation coefficient between min wt.% and FASA$_p$ (Figure 4E), ASA/ASA$^0$ (Figure S7B), R$_g$/R$_g^0$ (Figure S7C), and dipole moment (Figure S7D). We found only that FASA$_p$ is highly correlated to the min wt.%. Furthermore, FASA$_p$ is inversely correlated with min wt.%, indicating a higher polar fraction of solvent-accessible surface area from our simulation will experimentally form a stable nanosuspension at lower excipient:drug ratios. This result is easily rationalized, for more polar particles are more soluble in water and thus less likely to undergo destabilization from aggregation or Ostwald ripening [25].

While our modeling supports the hypothesis that a higher polar fraction of solvent-accessible surface area results in a more stable nanosuspension at lower concentrations of excipients, we still lacked a quantitative way to predict nanosuspension stability. To make our predictions more quantitative, we simplified our data by dividing it into two groups. In the first group we label as "successes", milling at certain excipient:drug ratios results in a stable nanosuspension. In the other group, which we label as "failures", milling does not result in a stable nanosuspension regardless of the excipient:drug ratio. Each group

consists of eight drug–excipient combinations (Table S3). We then used a logistic regression to calculate the probability of obtaining a "success" based on the $FASA_p$ and found

$$\log\left(\frac{P_f}{P_s}\right) = 6.14 - 26.1 * FASA_p, \tag{1}$$

where $P_f$ is the probability of not successfully forming a nanosuspension and $P_s$ is the probability of successfully forming a nanosuspension (Figure 4F, n = 16). To examine the accuracy of predictions from this logistic regression model, we recomputed our model using 100 independent rounds of 4-fold cross validation and found a classification accuracy of $0.74 \pm 0.21$, indicating that our model should be applicable to data beyond the training set. Our model has a negative coefficient in front of $FASA_p$, again suggesting that a higher polar fraction of solvent-accessible surface area is more likely to successfully form a stable nanosuspension.

## 4. Discussion

### 4.1. Simulated Polar Fraction of Solvent-Accessible Surface Area Successfully Predicts Experimentally Determined Nanosuspension Stability

Comparisons between our MD simulations and experiments on naproxen, indomethacin, and itraconazole suggest that a higher polar fraction of solvent-accessible surface area results in a more stable nanosuspension. To test this hypothesis, we studied compound GDC-0810, a former breast cancer candidate compound. GDC-0810 was selected due to its high hydrophobicity, which makes it poorly soluble in water. Furthermore, it can be crystalized in a neutral and ionic form (Figure 5A,B) [45]. As we hypothesized an ionic crystal such as the GDC-0810 potassium salt would be more polar than a neutral crystal, GDC-0810 offered an opportunity to examine the effect of polarizability on nanoparticle stability between different states of the same compound. For both the neutral and the ionic crystal, we performed simulations with the same six excipients used in our training data, plus Tween 80 (Figure 5C), resulting in 14 new simulations.

We immediately observed that the binding mode of the excipient varies greatly by the crystallization state of the drug. The atomistic contact maps find that SDS adsorbs to different regions of the neutral crystal than those of the ionic crystal (Figure 5D–F). This trend is seen with all excipient simulations (Figures S9 and S10). Similarly, molecular contacts vary greatly with the charge state of the drug. Most significantly, nearly all excipient molecules remain bound to the neutral crystal throughout the entire simulation, resulting in a very low probability of having 0 drug molecules per excipient. However, in the ionic crystal, a significant number of excipients are in solution, as indicated by a much higher frequency at 0 drug molecules per excipient (Figure 5G–J). These differences suggest that most excipients have a lower affinity for the ionic crystal than the neutral crystal. Similar trends for the drug molecules per drug (Figure S11), the drug molecules per excipient (Figure S12), and the excipient molecules per excipient (Figure S13) demonstrate that the ionic crystal and neutral crystal have different structural features.

Next, we aimed to connect knowledge of property differences within our training set to the example of GDC-0810. We first calculated the surface properties for both the ionic and neutral crystal (Figures 6 and S14). $FASA_p$ is much higher for the ionic crystal than the neutral crystal, indicating that the ionic crystal is more likely to form stable nanosuspensions. To quantitatively predict nanoparticle stability, we inserted the $FASA_p$ into Equation (1). Our model predicts that the ionic crystal was likely to form stable nanosuspensions with any of the surfactants tested, or even without any surfactant at all. Meanwhile, the neutral crystal may form a stable nanosuspension with more polar excipients, such as SOS or SDS (Table 1).

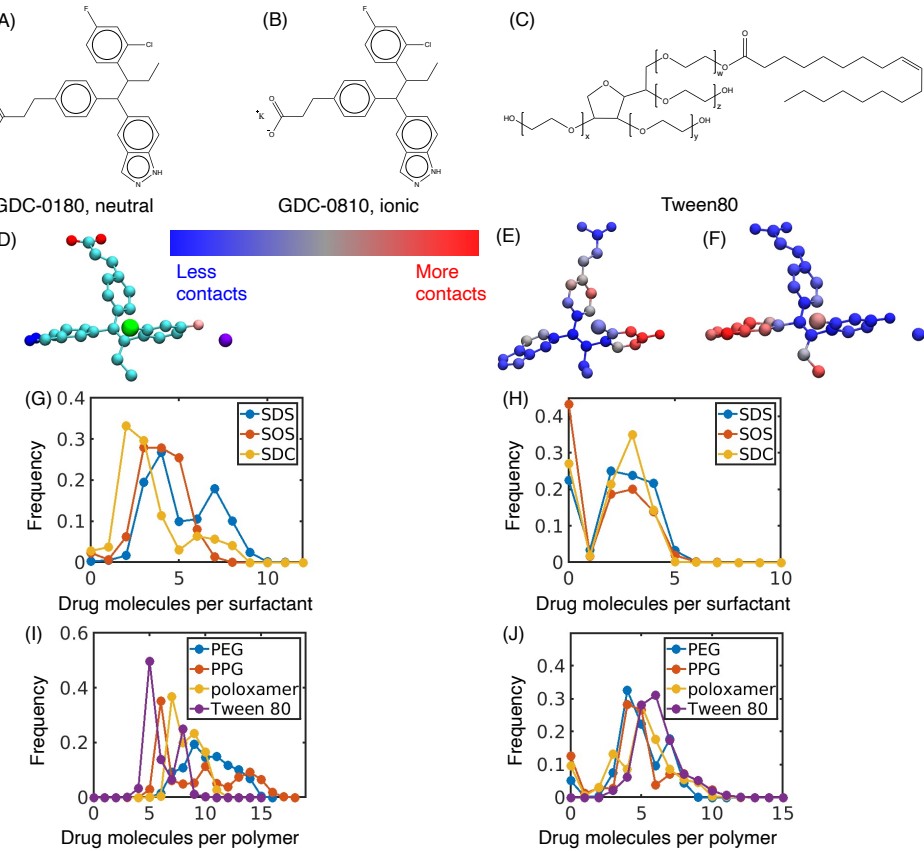

**Figure 5.** Summary of the GDC-0810 test performed in this study. Crystals of GDC-0810 were obtained in both neutral (**A**) and ionic (**B**) forms. Individual simulations were conducted with both crystal forms and each of the original 6 excipients (see Figure 1), as well as one additional excipient, Tween 80 (**C**), for a total of 14 simulations. In our simulations of Tween 80, we assumed w = x = y = z = 5. (**D**–**F**) Heavy atom contact maps for GDC-0810 with SDS. (**D**) Example structure of GDC-0810, colored by atom type (C–cyan, O–red, N–blue, F–pink, Cl–green, and K–purple). Contact map structures are given for the neutral crystal (**E**) and ionic crystal (**F**) of GDC-0810, where red highlights the most contacts with SDS and blue is the least contacts with SDS. (**G**–**J**) Frequencies of the number of drug molecules bound to a single excipient molecule during our simulations. Distributions are broken up between surfactant and polymer excipients for the neutral crystal (**G,I**) and ionic crystal (**H,J**).

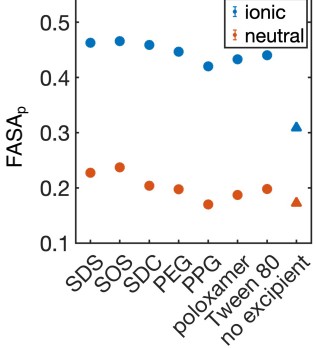

**Figure 6.** Fraction of polar surface area (FASA$_p$) for GDC-0810–excipient combinations. Ionic crystals of GDC-0810 (ionic, blue) are compared to the neutral crystals (neutral, orange). Circles indicate MD simulations with various excipients, while triangles indicate a single measurement on the crystal structure without excipients. Error bars represent the standard deviation of estimates from five independent time windows, are smaller than the symbols when not visible, and are only available from MD simulations.

**Table 1.** Probability of success ($P_s$) for GDC-0810 nanosuspension formulations, predicted by $FASA_p$ through the logistic regression in Equation (3). $FASA_p$ is extracted as an average from MD simulations for each excipient and directly from a single crystal structure in the case of no excipient (Figure 6).

| Excipient | Neutral $P_s$ | Ionic $P_s$ |
|---|---|---|
| SDS | 0.447 | 0.997 |
| SOS | 0.510 | 0.998 |
| SDC | 0.303 | 0.997 |
| PEG | 0.271 | 0.996 |
| PPG | 0.153 | 0.992 |
| Poloxamer | 0.221 | 0.994 |
| Tween 80 | 0.271 | 0.995 |
| no excipient | 0.161 | 0.871 |

Based on our MD simulations with different excipients, we hypothesized that ionic GDC-0810 may form nanosuspensions without excipients. Indeed, predictions of nanoparticle stability using the $FASA_p$ of the crystal without excipients (Figure 6) suggested that the GDC-0810 salt ($P_s = 0.871$), but not the protonated form ($P_s = 0.161$), is likely to form a stable nanosuspension. To evaluate this prediction, we utilized flash FNP [56] to create nanosuspensions of GDC-0810 salt. While different than the top-down milling techniques used by Ferrar et al. [39], FNP allowed us to control the pH of the solution, aiding in the formation of a sodium salt nanoparticle. Despite the differences between FNP and top-down milling, we hypothesized that nanosuspension stability is primarily dependent on the molecular characteristics of the drug and stabilizer, making our predictions transferable to this new system. Even without excipients, we found that the GDC-0810 salt formed nanosuspensions, which we characterized with dynamic light scattering. Our analysis revealed that the nanoparticles were stable, with an average size of 244.6 nm and % PD (percent polydispersity) of 26.8% upon formation (Figure 7A). Furthermore, despite an increase in polydispersity, the particles remained on the nanoscale after incubation for over 3 years and had an average size of 225 nm and % PD of 49.7% (Figure 7B). These results suggest that the salt nanosuspension would be a good choice of formulation for this poorly soluble compound.

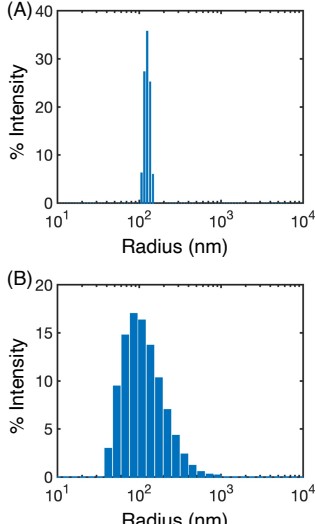

**Figure 7.** Stability of GDC-0810 salt nanoparticles measured by dynamic light scattering. These measurements were taken $T \approx 0$ years (**A**) and $T \approx 3.5$ years (**B**) after nanosuspension creation.

## 4.2. Leveraging Two-Dimensional Properties to Predict Nanosuspension Stability

The results with GDC-0810 support our previous theory that a higher polar fraction of solvent-accessible surface area results in a more stable nanosuspension. In principle,

one could run molecular dynamics simulations and test various combinations to discover the optimal excipient for a given drug. However, such simulations are computationally expensive, which motivated us to find underlying two-dimensional properties of drugs and excipients that can predict the simulated values of $\text{FASA}_\text{p}$.

We then hypothesized that hydrogen bond acceptors, hydrogen bond donors, and charged molecules contribute most to $\text{FASA}_\text{p}$. Using these properties for both the drug (Table S6) and excipient (Table S7), we used the least absolute shrinkage and selection operator (LASSO) to fit a linear regression model for $\text{FASA}_\text{p}$ with all systems we studied (Figures 4C and 6). LASSO is a technique that performs both variable selection and regression, improving the accuracy and interpretability of the resulting regression model (Figure S15). The model that minimizes the mean squared error (n = 32) takes the expression

$$\text{FASA}_\text{p}^\text{pred} = 13.9 \frac{HA_\text{drug}}{MW_\text{drug}} + 27.0 \frac{HD_\text{drug}}{MW_\text{drug}} + 2.3 \frac{HA_\text{exc}}{MW_\text{exc}}$$
$$+ 4.3 \frac{HD_\text{exc}}{MW_\text{exc}} + 118 \frac{|q_\text{drug}|}{MW_\text{drug}} + 10.6 \frac{|q_\text{exc}|}{MW_\text{exc}} - 0.108, \quad (2)$$

where $\text{FASA}_\text{p}^\text{pred}$ is the predicted value of $\text{FASA}_\text{p}$, $MW_\text{drug}$ is the molecular weight of the drug molecule, $HA_\text{drug}$ is the number of hydrogen bond acceptors in the drug molecule, $HD_\text{drug}$ is the number of hydrogen bond donors in the drug molecule, $q_\text{drug}$ is the charge of each drug molecule, $MW_\text{exc}$ is the molecular weight of the excipient molecule, $HA_\text{exc}$ is the number of hydrogen bond acceptors in the excipient molecule, $HD_\text{exc}$ is the number of hydrogen bond donors in the excipient molecule, and $q_\text{exc}$ is the charge of each excipient molecule. This model fits the calculated properties from simulations well, resulting in a high correlation between the 2-D property-predicted and simulated $\text{FASA}_\text{p}$ (Figure 8). Furthermore, hydrogen bond acceptors, hydrogen bond donors, and charge all contribute to a higher value of $\text{FASA}_\text{p}^\text{pred}$. Interestingly, we also observe that hydrogen bond donors contribute more than hydrogen bond acceptors, a common trend in drug discovery settings [57]. This model suggests that excipients with more hydrogen bond donors, hydrogen bond acceptors, and net charge will result in more stable nanosuspensions. Finally, the charge state of drug molecules contributes significantly to $\text{FASA}_\text{p}^\text{pred}$, which suggests that salt forms of drugs are more likely to form stable nanosuspensions.

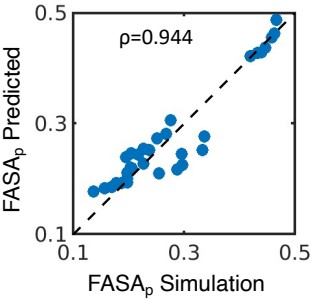

**Figure 8.** Predicting the $\text{FASA}_\text{p}$ from 2-D molecular properties. Predicted values of $\text{FASA}_\text{p}$ are compared to those from the simulation, with $\rho$ representing the Pearson correlation between the two datasets.

To test whether our model of two-dimensional properties is capable of predicting nanoparticle stability, we applied the same tests on our $\text{FASA}_\text{p}^\text{pred}$ that we applied on $\text{FASA}_\text{p}$. First, we examined the correlation between $\text{FASA}_\text{p}^\text{pred}$ and the min wt.% (Figure 9A). Unlike the simulated values of $\text{FASA}_\text{p}$, the $\text{FASA}_\text{p}^\text{pred}$ shows little correlation with the min wt.%. This deviation could originate from a variety of factors, such as the binding mode of a given excipient or crystal size and shape.

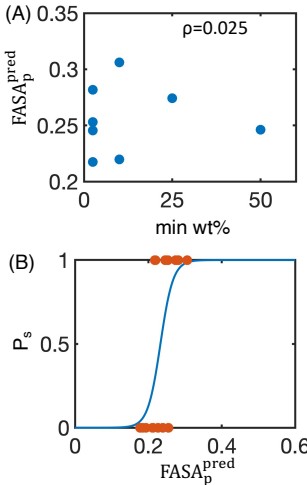

**Figure 9.** Determining nanoparticle stability using the the fraction of polar surface area predicted from two-dimensional properties ($\mathrm{FASA_p^{pred}}$). (**A**) $\mathrm{FASA_p^{pred}}$ plotted against the minimum weight percentage of excipient needed to form a stable nanosuspension (min wt.%). The Pearson correlation coefficient ($\rho$) between $\mathrm{FASA_p^{pred}}$ and min wt.% is also shown. (**B**) Using $\mathrm{FASA_p^{pred}}$ to predict the probability that a stable nanosuspension is formed ($P_s$) using a logistic regression. True data (orange dots) and the resulting fit (blue line) are both shown.

Despite our modeled $\mathrm{FASA_p^{pred}}$ not correlating with min wt.%, it does show some success at predicting nanosuspension stability. Following our procedure for $\mathrm{FASA_p}$, we labeled a drug–excipient combination as a "success" if it formed a stable nanosuspension upon milling at certain excipient:drug ratios and labeled it as a "failure" if it did not form stable nanosuspensions upon milling at any excipient:drug ratios. For these binary data, we performed logistic regression to calculate the probability of getting a "success" based on the $\mathrm{FASA_p}$ and found

$$\log\left(\frac{P_f}{P_s}\right) = 13.5 - 58.0 * \mathrm{FASA_p^{pred}}, \tag{3}$$

where $P_f$ is the probability of not successfully forming a nanosuspension, and $P_s$ is the probability of successfully forming a nanosuspension (Figure 9B, n = 16). Similar to our previous regression model in Equation (1), we recomputed this updated model using 100 independent rounds of 4-fold cross validation and found a nearly identical classification accuracy of $0.74 \pm 0.20$, indicating that our model should be applicable to data beyond the training set. The expression itself is also similar to that found in Equation (1) and suggests that higher values of $\mathrm{FASA_p^{pred}}$ result in more stable nanosuspensions. Finally, we apply Equation (3) to predict the probability of "success" for GDC-0810 nanoparticles (Table 2). Our model predicts that any excipient would be able to form a stable nanosuspension with the salt form of GDC-0810, which aligns with our experimental observation that the salt of GDC-0810 can form a nanosuspension without excipient.

Overall, our model suggests that drug–excipient combinations with a higher polar fraction of solvent-accessible surface area will form the best nanosuspensions. This property can be maximized by increasing the charge and hydrogen bonding properties of excipients or by using salt forms of drug molecules. However, this simplified model should be used with some amount of caution. First, to test the ability of excipients to adsorb to stable nanoparticles, we placed restraints on the position of drug atoms. These extra potentials prevent crystal rearrangements, which would be overestimated in unrestrained simulations, because computational feasibility required using smaller nanoparticles than those predicted experimentally. As such, effects arising from crystal deformation may not be fully captured by our model. Further, it is possible that a sufficiently polar drug–excipient combination will be soluble in solution and not form a stable nanosuspension. We begin to observe such

trends with the ionic form of GDC-0810, where less excipients adsorb to the drug crystal in our simulations. These effects may also manifest themselves in drug solubility, where potentially soluble salt isoforms may prevent the formation of stable nanosuspensions. Such considerations are not taken into account in our model, which may mean predictions from Equations (1) and (3) might overestimate the stability of some nanosuspensions. As our work here primarily addressed explored hydrophobic drug–excipient combinations in which aggregation and Ostwald ripening are the primary concerns, future studies may want to employ similar methods to test increasingly polar excipients and elucidate the balance between stable nanosuspensions and solubility.

**Table 2.** Probability of success ($P_s$) for GDC-0810 nanosuspension formulations, predicted by $FASA_p^{pred}$ through the logistic regression in Equation (3). In the case of no excipient, the data are not used for fitting the LASSO, and all terms involving excipients are set to 0.0.

| Excipient | Neutral $P_s$ | Ionic $P_s$ |
|---|---|---|
| SDS | 0.438 | 1.000 |
| SOS | 0.761 | 1.000 |
| SDC | 0.336 | 1.000 |
| PEG | 0.141 | 1.000 |
| PPG | 0.064 | 1.000 |
| Poloxamer | 0.091 | 1.000 |
| Tween 80 | 0.097 | 1.000 |
| no excipient | 0.004 | 1.000 |

## 5. Conclusions

In this study, we utilized molecular dynamics simulations to evaluate the characteristics that determine nanosuspension stability. We began by examining how hydrophobic interactions drive how excipients adsorb to naproxen, indomethacin, and itraconazole. Then, we showed that the polar fraction of solvent-accessible surface area from our simulated drug–excipient nanoparticle was correlated with the minimum ratio of excipient:drug necessary to form stable nanosuspensions. Inspired by this relationship, we created a logistic regression model that predicts the probability of forming a stable nanosuspension from the polar fraction of solvent-accessible surface area from simulation. To test this model, we studied GDC-0810, which can be crystallized in different isoforms. Our simulations revealed that the crystal isoform had a large impact on excipient adsorption. When we studied the fraction of polar surface area, we found that the ionic crystal form is far more polar, and, therefore, is more likely to form stable a stable nanosuspension than the neutral crystal. These results aligned with experimental observations where we observed stable nanosuspensions of GDC-0810 without any excipient. We were then able to extend our model to two-dimensional properties and discovered our trends can be explained by hydrogen bonding and charge properties. While this simplified model was unable to resolve the more subtle differences given by the minimum weight percentage, it could predict the probability of forming a successful nanosuspension. Overall, we utilized a combination of modeling and targeted experiments to discover that salt forms of drug crystals and more polar excipients can improve nanosuspension stability. Future studies may want to expand our model by increasing the coverage of excipient chemical space, including more polar compounds, as well as aliphatic or aromatic polymers.

**Supplementary Materials:** The following supporting information can be downloaded at: https://www. mdpi.com/article/10.3390/pharmaceutics16010050/s1, Figure S1: Images of the final configurations for all training set simulations; Figure S2: Heavy atom contact maps for where indomethacin molecules come in contact with excipient molecules; Figure S3: Heavy atom contact maps for where itraconazole molecules come in contact with excipient molecules; Figure S4: Frequency of the number of drug molecules bound to a single drug molecule during our simulations; Figure S5: Frequency of the number of excipient molecules bound to a single drug molecule during our simulations; Figure S6:

Frequency of the number of excipient molecules bound to a single excipient molecule during our simulations; Figure S7: Evidence that other surface properties do not serve as good of predictors of nanoparticle stability as FASA$_p$; Figure S8: Images of the final configurations for all GDC-0810 simulations; Figure S9: Heavy atom contact maps for where neutral GDC-0810 molecules come in contact with excipient molecules; Figure S10: Heavy atom contact maps for where ionic GDC-0810 molecules come in contact with excipient molecules; Figure S11: Frequency of the number of drug molecules bound to a single drug molecule during GDC-0810 simulations; Figure S12: Frequency of the number of excipient molecules bound to a single drug molecule during GDC-0810 simulations; Figure S13: Frequency of the number of excipient molecules bound to a single excipient molecule during GDC-0810 simulations; Figure S14: Additional surface properties for GDC-0180 crystals with various excipients; Figure S15: Fitting FASA$_p$ from 2-D molecular properties with the least absolute shrinkage and selection operator (LASSO); Table S1: Additional experimental characterization of naproxen, indomethacin, and itraconazole nanosuspensions; Table S2: Minimum weight percentage of excipient:drug needed to form a stable nanosuspension (min wt.%) based on experimental characterization; Table S3: Summary of experimental results used for our logistic regression model; Table S4: Number of drug and excipient molecules used in creating our training set; Table S5: Number of drug and excipient molecules used in creating our test set; Table S6: 2-D properties of drug molecules used for fitting in Figure 8; Table S7: 2-D properties of excipient molecules used for fitting in Figure 8.

**Author Contributions:** Conceptualization, A.P.L., E.S.L., B.D.S., and D.H.L.; methodology, A.P.L., E.S.L., B.D.S., and D.H.L.; formal analysis, A.P.L.; investigation, A.P.L., E.S.L., and D.H.L.; data curation, A.P.L.; writing—original draft preparation, A.P.L.; writing—review and editing, A.P.L., E.S.L., B.D.S., and D.H.L.; visualization, A.P.L.; supervision, E.S.L., B.D.S., and D.H.L. All authors have read and agreed to the published version of the manuscript.

**Funding:** This study was funded by Genentech, Inc.

**Institutional Review Board Statement:** Not applicable.

**Informed Consent Statement:** Not applicable.

**Data Availability Statement:** Data available upon request.

**Conflicts of Interest:** All authors were employed by the company Genentech, Inc. when the work was performed. The authors declare that this study received funding from Genentech, Inc. The funder was not involved in the study design, collection, analysis, interpretation of data, the writing of this article or the decision to submit it for publication.

## Abbreviations

The following abbreviations are used in this manuscript:

| | |
|---|---|
| MD | Molecular dynamics |
| FNP | Flash nano precipitation |
| SDS | Sodium dodecyl sulfate |
| SOS | Sodium octyl sulfate |
| SDC | Sodium deoxycholate |
| PEG | Polyethylene glycol |
| PPG | Polypropylene glycol |
| FASA$_p$ | Fraction of polar surface area |
| ASA | Accessible surface area |
| R$_g$ | Radius of gyration |
| LASSO | Least absolute shrinkage and selection operator |
| MIVM | Multi-inlet vortex mixer |
| min wt.% | Minimum weight percentage of excipient needed to form a stable nanosuspension |
| % PD | Percent polydispersity reported as the polydispersity divided by the estimated hydrodynamic radius of the particle size population multiplied by 100 |

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
