# Peer review of "Utilizing Molecular Simulations to Examine Nanosuspension Stability"

_pharmaceutics, doi:10.3390/pharmaceutics16010050_

Round 1

Reviewer 1 Report

Comments and Suggestions for Authors

Recommendation: Accept with major edits

Notes, general: This study aims to develop a molecular dynamics model that connects experimentally-observed colloidal stability with physical chemical properties of a hydrophobic drug and excipient stabilizer. The resulting tool is interesting and may be practically useful, especially if it can be expanded to solvent systems beyond 100% water. The experimental example at the end with GDC-0810 is not well motivated with the rest of the work and should be presented differently or moved to the SI. Some of the results of that test (especially the three year stability) need to be explained better. The study also suffers from some vague and inappropriate language and should be proofread for this. Overall I think it is interesting and will be ready for acceptance after a round of significant revisions.

Notes, specific: 

- Line 52 “expeirmental” typo

- Line 82 “Camridge” typo

- Line 94 “excpients” typo

- Line 121: The rationale for the choice of streams into the MIVM was not clear. If the drug is dissolved in 0.4 eq. NaOH, is it drug expected to precipitate at 0.13 eq. after dilution / mixing? Manufacturers state that the drug is not soluble in water, and its logP is very high (above 7), so how were you able to dissolve it at 15 mg/mL in water? What is the final pH after mixing? Were the other streams deionized water, or did they also contain NaOH? Was a surfactant or stabilizer added to either any of the non-drug water streams or to the collection bath? Why was only GDC-0810 chosen for flash nano precipitation?

- It is also not clear why flash nano precipitation is being used. This study appears to be a sequel to Ferrar et al., who used a resonant acoustic mixing as a top-down approach to make nanocrystals. Flash nano precipitation typically involves the use of organic solvents, which are not accounted for in the molecular dynamics simulation (the system studied is all water) nor used in the experimental section.

- It would have been nice to compare an aromatic polymer excipient (e.g polystyrene-b-PEG) versus an aliphatic polymer excipient (e.g. polycaprolactone-b-PEG) in this, especially since several of the studied drugs are highly aromatic.

- Line 197 “we wondered if we could relate our findings” tone is not appropriate

- Line 243 It is unclear why “ionic form” is used here and elsewhere instead of “potassium salt,” which is the structure shown in figure 5.

- Line 266 “Our model predicts that the ionic crystal was likely to form stable nanosuspensions under any conditions, while the neutral crystal may form a stable nanosuspension with more polar excipients like SOS or SDS” What does ‘under any conditions’ mean? Is there a required pH range or temperature range? The language is too vague. If you mean ‘with any of the surfactants tested, or even without any surfactant at all,’ say that instead of ‘under any conditions.’

- There needs to be a clearer connection between the stated goal of this work from the abstract / introduction and the GDC-0810 example. GDC-0810 is strongly hydrophobic, but this is not stated anywhere in the text so far. Your useful simulation tool correctly predicted that the ionized form of GDC-0810 can self-stabilize nanocrystals of itself without an additional stabilizer, which is interesting, but the model was built using data from milling – so why is the example using nanoprecipitation?

- Tables 1 and 2 should include a row of ‘no stabilizer’

- Is the DLS result in Figure 7 at t=0, or after t=3 years? The PDI increasing to 0.49 is concerning, and you need to show both particle size distributions side-by-side before you can claim stability as you have.

- The equation between lines 295 and 296 is cut off.

Comments on the Quality of English Language

See above.

Reviewer 2 Report

Comments and Suggestions for Authors

I read with interest the article entitled Utilizing molecular simulation to examine nanosuspension stability

The article is very well written and addresses an important problem: the stability of nanoemulsions.

The only critical note concerns the Introduction section. I believe that in the introduction the authors should focus on the goal and not the results and this should be corrected. A summary of the results should be included more in the Conclusions section.

The authors talk about "Nanoparticles with a diameter of 225 nm were prepared" - please clarify which of the parameters is below 100 nm - only then we are talking about the nano scale

Reviewer 3 Report

Comments and Suggestions for Authors

The submitted manuscript "Utilizing molecular simulation to examine nanosuspension stability" presents a well-structured and coherent research article that delves into the use of drug nanosuspensions to enhance the bioavailability of poorly soluble drugs. The abstract clearly highlights the key challenges in formulating these nanosuspensions, such as the need for stabilizing excipients and the lack of a clear rationale for excipient selection. The use of molecular dynamics simulations to investigate drug-excipient interactions for various combinations adds a valuable dimension to the research.

The identification of hydrophobic interactions as the driving force for excipient adsorption to drug nanoparticles provides a clear molecular insight into the formulation process. Additionally, the correlation between the fraction of polar surface area and nanosuspension stability serves as a promising predictor for experimental measurements, contributing to the practical application of the research findings. The  application of the model to the uncharacterized drug compound GDC-0810 and the successful prediction of more stable nanosuspensions with salt forms underscore the practical utility of the proposed computational approach.

I found the manuscript to be well written and presented in a visually appealing manner with appropriate graphical images to explain theory and experimental data.

In Figure 1B I would consider adding some of the description to the image by labelling the different coloured parts directly which will also have the benefit of reducing the amount of text in the Figure caption

Comments on the Quality of English Language

General English style is excellent with only minor spell check required

For example, in the sentence Line 52 "To rationalize these previous expeirmental results," "expeirmental" should be corrected to "experimental."
